# “Moving Forward with Life”: Acceptability of a Brief Alcohol Reduction Intervention for People Receiving Antiretroviral Therapy in South Africa

**DOI:** 10.3390/ijerph17165706

**Published:** 2020-08-07

**Authors:** Bronwyn Myers, Charles D. H. Parry, Neo K. Morojele, Sebenzile Nkosi, Paul A. Shuper, Connie T. Kekwaletswe, Katherine R. Sorsdahl

**Affiliations:** 1Alcohol, Tobacco and Other Drug Research Unit, South African Medical Research Council, Tygerberg 7505, South Africa; cparry@mrc.ac.za (C.D.H.P.); neo.morojele@mrc.ac.za (N.K.M.); sebenzile.nkosi@mrc.ac.za (S.N.); connie.kekwal@gmail.com (C.T.K.); 2Department of Psychiatry & Mental Health, University of Cape Town, Cape Town 7925, South Africa; 3Department of Psychiatry, Stellenbosch University, Tygerberg 7505, South Africa; 4Department of Psychology, University of Johannesburg, Johannesburg 2006, South Africa; 5Institute for Mental Health Policy Research, Centre for Addiction and Mental Health, Toronto, ON M5T1R8, Canada; pau.shuper@uconn.edu; 6Alan J. Flisher Centre for Public Mental Health, Department of Psychiatry & Mental Health, University of Cape Town, Cape Town 7701, South Africa; katherine.sorsdahl@uct.ac.za

**Keywords:** South Africa, alcohol reduction, anti-retroviral therapy, motivational interviewing, problem-solving therapy, HIV, global mental health

## Abstract

Background: In South Africa, interventions are needed to address the impact of hazardous drinking on antiretroviral therapy among people living with HIV (PLWH). Participant feedback about these interventions can identify ways to enhance their acceptability. We interviewed participants in a randomized controlled trial of a brief motivational interviewing and problem-solving therapy (MI-PST) intervention about their perceptions of this alcohol-reduction intervention. Methods: The trial was conducted in HIV treatment clinics operating from six hospitals in the Tshwane region of South Africa. We conducted qualitative in-depth interviews with a random selection of participants. Twenty-four participants were interviewed after the final intervention session and 25 at the six-month follow up. Results: Participants believed that it was acceptable to offer PLWH, an alcohol reduction intervention during HIV treatment. They described how the MI-PST intervention had helped them reduce their alcohol consumption. Intervention components providing information on the health benefits of reduced consumption and building problem-solving and coping skills were perceived as most beneficial. Despite these perceived benefits, participants suggested minor modifications to the dosage, content, and delivery of the intervention for greater acceptability and impact. Conclusions: Findings highlight the acceptability and usefulness of this MI-PST intervention for facilitating reductions in alcohol consumption among PLWH.

## 1. Introduction

Despite an extensive HIV prevention and treatment program, South Africa continues to have one of the world’s largest HIV epidemics. The most recent national HIV survey estimates that 14% of all South Africans are now living with HIV [1]. To prevent the onward transmission of HIV, South Africa subscribes to UNAIDS’ 90–90–90 strategy, namely that 90% of the South African population should be tested for HIV, 90% of individuals who test positive should receive sustained antiretroviral therapy (ART), and 90% of individuals on ART must attain viral suppression [2]. South Africa has not met these targets, with the most recent estimates suggesting that 85% of adult South Africans know their HIV status, 71% of people living with HIV (PLWH) are on ART, and 87% of PLWH on ART are virally suppressed [1]. Hazardous alcohol use among people living with HIV (PLWH) who use ART is one threat to the attainment of these targets. Systematic reviews have demonstrated that hazardous alcohol use is associated with poorer adherence to ART, higher viral load, and greater likelihood of treatment failure and early death [3,4,5,6]. These findings highlight the importance of identifying and addressing hazardous alcohol use among PLWH.

The alcohol and HIV syndemic is well-recognized in South Africa [7] and elsewhere in Southern Africa [8,9], where problem drinking seems to be concentrated in populations and communities most affected by HIV. Rates of hazardous alcohol use are high in South Africa [10], where about 50% more alcohol is consumed than in the rest of Africa [11]. Although only a third (31%) of South Africans report consuming alcohol in the past year, heavy episodic drinking is the norm amongst those that do drink—71% of men and 34% of women report this pattern of drinking [11]. The average drinker in South Africa consumes an estimated 30 L of absolute alcohol per year, placing South Africa in the top six countries globally for amount of alcohol consumed per drinker [11]. In South Africa, studies have demonstrated high rates of heavy episodic drinking among PLWH who drink [12,13,14]. Given the impact of hazardous alcohol use on virologic control, there is a clear need to routinely screen PLWH on ART for hazardous alcohol use and to provide alcohol-reduction interventions among those who screen positive. A recent systematic review of 21 trials (8461 PLWH) provides evidence that brief interventions are effective for reducing the frequency of alcohol use, with interventions that focused exclusively on alcohol seemingly more effective than those that addressed alcohol as part of a multi-faceted HIV behavior change intervention [15]. However, as only four trials in this review focused exclusively on alcohol, there is a need for additional comparative efficacy research of alcohol-focused interventions for PLWH [15]. In response to this need, we tested the efficacy of an alcohol-focused intervention for reducing alcohol consumption and improving ART adherence and HIV treatment outcomes among PLWH in South Africa [16].

To better understand participants’ responses to and interactions with this intervention, we examined patients’ experiences and perceptions of the intervention. Understanding patients’ perceptions of the intervention may help explain how patients engage and apply the intervention material, for whom it works, and the context in which it is most effective. This can guide modifications to the intervention necessary for enhancing its appeal and impact [17]. More specifically, this study aimed to explore participants’ perceptions of the acceptability of the structure and content of the intervention and its perceived usefulness for facilitating behavioral change. We hoped that this evaluation would identify ways in which to modify the structure and content of the intervention to better address the context and therapeutic needs of PLWH who drink at hazardous or harmful levels.

## 2. Materials and Methods

This qualitative process evaluation is nested within a larger randomized trial of an alcohol reduction intervention. The methods used in the trial are described in detail elsewhere [16].

Participants were recruited from outpatient ART clinics operating within four secondary and two tertiary hospitals within the Tshwane district, in the Gauteng province of South Africa. Participants were recruited into the trial if they reported being HIV-positive; on ART for at least three months; at least 18 years of age; not currently on treatment for tuberculosis; not having participated in the formative phase of the study; living in Tshwane or within the clinic’s catchment area; and who were hazardous or harmful drinkers, based on their Alcohol Use Disorders Identification Test-Consumption (AUDIT-C) scores (≥3 for women and ≥4 for men).

The trial recruited 623 participants who completed baseline behavioral assessments that included the full AUDIT and other questions regarding patterns of alcohol use [18]. Thereafter, participants were randomly assigned to either the intervention condition (*n* = 305) or a treatment-as-usual comparison condition (*n* = 318). Participants assigned to the intervention condition were offered an intervention that combined motivational interviewing (MI) and problem-solving therapy (PST). The intervention comprised four modules of MI-PST delivered over two sessions that were spaced a week apart. The hypothesis was that maladaptive problem solving and inability to cope with stress underpinned hazardous alcohol use [19,20]. This hypothesis emerged from formative work which explored factors underpinning alcohol use among PLWH and its impact on ART adherence as well as openness to and preferences for alcohol reduction interventions [21,22]. The intervention included motivational elements to build readiness for alcohol behavior change and PST content to help patients cope with life stressors, deal with negative emotions, and accept problems that cannot be solved without relying on alcohol as an avoidant coping strategy [23,24]. There is evidence that the combination of these two approaches is acceptable and effective for reducing hazardous alcohol use in South African patient populations [25,26,27,28,29].

In this study, trained counsellors delivered individual counselling in a private room next to the ART clinic. During the intervention, the counsellor provided feedback on the participants’ risk for alcohol-related harms, helped the participant set goals and identify barriers to change, and guided the participant in identifying life problems that may contribute to alcohol use while teaching the participant a structured approach to resolving these problems. Participants learned strategies for addressing problems that are important and resolvable, for managing negative thoughts, and for coping with important problems that are unresolvable. Table 1 describes the counselling modules, with each session building on the skills learned in the previous modules. All modules included opportunities to apply newly learned skills through exercises and home-based activities. These activities were contained in a patient handbook that also summarized the module’s content. From enrolment, participants had four weeks within which to complete these sessions. Of the 305 participants assigned to the intervention, 225 (74%) completed all four modules.

To obtain participants feedback about their experiences of the intervention, we conducted qualitative in-depth interviews at two time-points: on completion of the last intervention session and again at the six-month study endpoint. We chose these two time-points as we thought views of the mode and style of intervention delivery would be more salient immediately after completion of the intervention, while we would only obtain information about whether participants had been able to apply the skills they had learned and intervention components that facilitated change at the six-month follow-up interview.

At each of these time-points, 12% of the 305 participants who were assigned to the intervention condition at baseline were randomly selected for an interview: 36 unique participants at the post-intervention time-point and 37 unique participants at the six-month endpoint. We interviewed 49 (67%) of the 73 participants initially selected for an interview: 24 of the 36 participants selected for a post-intervention interview and 25 of the 37 participants selected for an in-depth interview at the six-month follow-up appointment. Of the 24 selected participants that were not interviewed, 14 (58%) were lost to follow-up, 8 (33%) had withdrawn from the study, and 2 (8%) did not want to participate in an in-depth interview. There were no baseline differences between participants who were interviewed and those who were selected, but not interviewed (Table 2).

The interviews were conducted by female research assistants with postgraduate qualifications, training in qualitative research methods, experience in conducting interviews, and who were proficient in the local languages. The research assistants were not known to the participants prior to the interviews. The study was explained in participants’ first language and written informed consent was obtained before conducting the interview. The research assistants used an interview guide (containing opening questions and follow-up probes) to structure these interviews which were conducted in Setswana or English, the main languages spoken in the region. Similar questions were asked at each time point. Questions explored what participants remembered from the intervention, their experiences of counselling, the extent to which they had applied the information and skills they had learned to life problems, components of the intervention that had helped facilitate change, and their recommendations for improving the content, format, and delivery of the intervention. Interviews lasted up to 45 min, were audio-recorded and transcribed verbatim. Participants were provided with refreshments, a grocery voucher to thank them for their time, and their transport costs associated with attending these interviews were reimbursed. The interviews occurred in private rooms within the ART clinics that operated as research sites.

Ethical approval for the study was granted by the Research Ethics Committee of the South African Medical Research Council (EC003-2-2014 in February 2014). Permissions for the study were obtained from the hospitals, the health districts, and the Gauteng Provincial Department of Health. The trial was registered with the Pan African Clinical Trials Register (PACTR201405000815100, date of registration: 22 April 2014).

The framework approach [30] was used for qualitative data analysis. This approach is increasingly used in public health research as it allows for deductive types of qualitative analysis where data are captured under specific themes required to answer the research questions, while also allowing for inductive analyses where new themes emerge from the data [31]. The first and last authors conducted the initial process of familiarization with the data through review of transcripts. These authors developed the initial coding framework and individually coded the first five transcripts. Following this, they discussed coding, major overarching themes and sub-themes and adjusted the coding framework. Coding then continued independently. Although all the transcripts were coded, no new codes emerged after coding a third of the transcripts, suggesting thematic saturation. Any coding disagreements were resolved through discussion. A third person was not needed to break coding ties. While analyzing the data, the researchers remained aware of their personal beliefs and assumptions about the utility of the intervention and its limitations.

## 3. Results

The 49 participants who were interviewed were mainly women (67%) and were 41 years of age (SD = 9.3) on average. Three quarters of the sample reported significant symptoms of depression (76%) and their average AUDIT score was 8.6 (SD = 4.6), indicating hazardous alcohol use. Table 2 provides a description of the demographic and clinical characteristics of the study participants as well as the trial participants who were selected for this qualitative study but did not participate.

Three themes emerged from the interviews that reflect participants’ perceptions of the acceptability and usefulness of this intervention. Similar themes emerged for participants interviewed post-intervention and at the six-month follow-up (6MFU), although participants at the 6MFU provided a richer description of the usefulness of the intervention and were able to provide concrete examples of how they had applied specific skills learned during the intervention to facilitate behavioral change. Participants at the post-intervention point would not have had an opportunity to apply skills learned during module three and four of the intervention. The first theme describes participants’ perceptions of the acceptability of screening and brief alcohol-focused interventions for patients living with HIV. The second theme describes participants’ views of the usefulness of the intervention for reducing alcohol use and addressing life stressors. The third theme reflects participants’ views on how the intervention could be modified for greater reach and impact. These themes are described below and are illustrated with quotes.

### 3.1. Alcohol-Focused Interventions were Deemed Acceptable and an Essential Adjunct to Clinical Care

When asked for feedback about the process of being screened for hazardous alcohol use, almost all participants felt that screening helped them understand the level of risk associated with their current pattern of alcohol use. Those participants that initially expressed ambivalence towards this screening did not view their drinking as problematic:
I did not see that I had a problem with alcohol. It (the screening) was able to open my eyes to see that I have a problem. (Male PID 7, post-intervention)
I wasn’t even aware that I have a drinking problem, I only became aware when I started talking about it. (Female PID 70; 6MFU)

Participants agreed that the offer of a brief alcohol reduction intervention was largely acceptable to them. Many of them reported feeling stressed and overwhelmed with life problems and perceived the offer of counselling as a rare opportunity to discuss the problems in their lives. This is reflected in the following comments:
When I was approached, I thought it [the counselling] was going to give me light and make sense to me, show me what I should do in life. (Female PID 22; post-intervention)
I have many hardships and I wanted help. With the situation that I am in, I wondered what could help me? So I saw this programme as something that was for me. (Male PID 31; 6MFU)

Several participants described how the attitude and qualities of the counsellor enhanced the extent to which they viewed the offer of an alcohol-reduction intervention as acceptable. They described how the opportunity to talk to a counsellor who was knowledgeable about alcohol use and provided a safe, confidential, and non-judgmental space to discuss problems and concerns was highly valued.

My counsellor spoke to me in a friendly manner and listened to me when I talked. She would go extra lengths for me. I liked the way she listened to me. (Female PID 54; 6MFU)

She is a person who can understand a situation and then at the end help come up with a solution, she is a friendly person. She doesn’t judge, she listens. (Male PID 7; post-intervention)

In several instances, participants described the intervention as an essential adjunct to the clinical care they received at the facility. Participants remarked how their HIV providers only addressed their physical and HIV disease-related concerns, with little attention to emotional and social issues that impacted on their well-being. As a result, participants tended to perceive the intervention as addressing a gap in current services:
I saw this counselling as very important because of the way they have time. Here at the clinic, they don’t have that time. I am sorry to say this but here they (nurses) are only rushing so that the line can move, finish and then go home. (Male PID 16; post-intervention)
It was helpful because we don’t get much counselling and sometimes I feel that when you have a problem, you don’t talk to them (nurses), you just come in, they sign your file, you take your treatment and you go home… so it’s like I am getting treatment that side and advice about my life this side. (Female PID 56; 6MFU)

When asked specifically, all participants said that they would recommend the intervention to other people living with HIV who use alcohol, and several suggested it would be suitable for all people living with HIV regardless of whether they use alcohol. As one participant reflected:
I think this program is good for any person taking ARV medication because this counselling has really helped me a lot. It is good to talk to someone who is trained. (Female PID 29; 6MFU)

### 3.2. Usefulness of the MI-PST Intervention for Facilitating Alcohol Reduction and Other Behaviour Change

Almost all participants thought the intervention “was helpful” for helping participants reduce their alcohol use, improve their emotional well-being and make other positive changes to their lives. When describing the usefulness of the intervention, participants commented:
Since I came here for counselling, I feel like a lot has changed, even my mind is functioning better. (Male PID 29; post-intervention)
I feel lighter because I was always feeling pain in my spirit and remember when I started coming here I was very hurt… and as time went I got better. (Female PID 38; 6MFU)

Participants seemed to value the psychoeducation material contained in the intervention which provided them with feedback about their personal risks for alcohol-related harms and psychoeducation about alcohol use and health. Several participants articulated that prior to receiving the intervention they were not “aware that a lot of alcohol is not advised when one is taking treatment” and that they only “started knowing it after being part of this study.” Once they learned about the health risks associated with their current level of alcohol consumption, they felt motivated to reduce their alcohol use in order to remain healthy. As one participant reflected:
Sometimes you do things and you are not aware that they are wrong. I realise that if I am taking ARV’s I am not supposed to have problems or stresses, or drink too much alcohol. (Female PID 71; 6MFU)

For many participants, the information on standard drinks and container sizes was particularly salient and helped them quantify the amount of alcohol they were consuming. This information helped them manage the quantity of alcohol they consumed on any one occasion:
So now when I drink alcohol, I am careful because I look at the fact that I am drinking 750 mL that means I am drinking 2.2 beers. I didn’t know, I thought 750 mL (bottle) equals one drink. It taught me that sometimes we drink a lot of alcohol without being aware. (Male PID 49; 6MFU)

Several participants reported using the patient handbook (which contained goal setting activities and a drinking diary) to help them keep track of the amount of alcohol they were consuming. These participants experienced the handbook as a useful platform to support self-monitoring of alcohol intake and self-evaluation of progress towards their alcohol reduction goals. As one participant commented:
What I liked about this book is that I can use it as my diary, where I keep track of my drinking habit. I write down and calculate the percentage of alcohol whenever I have a drink and see the total. (Male PID 17; post-intervention)

The PST content of the intervention, contained in modules 2–4, also appeared to be valued by participants who agreed that the structured problem-solving approach taught during counselling had given them new skills for resolving everyday problems. Participants reflected that the PST approach had taught them to face their problems directly without having to turn to alcohol. As the following participants reflected:
I now accept any problem that comes my way and I can now solve problems. I don’t bottle them inside anymore, but I deal with them head on. (Female PID 54; 6MFU)
I see life with a different eye. I take things as they come and know how I should tackle them (problems). When I come across problems I take things step by step. I no longer rush to drink when I have stress. (Female PID 49; 6MFU)

Several participants also reflected on the usefulness of the PST content that focused on strategies for accepting and managing problems that cannot be changed. This seemed particularly salient for participants who were struggling to accept their HIV diagnosis and were using alcohol to cope with the negative feelings associated with this diagnosis:
I learned to accept myself in this situation that I am in… to accept that I will take ARVs forever. I could not accept it in the beginning, now I have accepted.(Male PID 23; post-intervention)

At the six-month endpoint, many participants provided concrete examples of how they had applied their newly acquired skills to resolve some of the problems in their lives that triggered or contributed to excessive alcohol use. Participants described using these skills to manage a range of life problems, including unemployment (with a few reporting that they had found part-time employment or started income-generation activities), relationship difficulties and interpersonal conflicts. These were salient life problems for both men and women. PST skills seemed particularly helpful for aiding the regulation of negative emotions such as anger, anxiety, and despair:
I was a person who had anger… I would turn small issues into big ones and I was quick to get angry. Since I have spoken to the counsellors, they have advised me what to do. I tried to do what they said and now I can see that at least I am not too quick to get angry. (Female PID 67; 6MFU)
In counselling, they taught me how to cope with negative thoughts. I am maybe not going to allow this thought to consume my mind. I usually have stress when I am alone and I think about a lot of things.(Female PID 71; 6MFU)

All participants interviewed at the six-month timepoint reported applying these PST skills to support their efforts to reduce their alcohol consumption. Although a few participants reported that they had stopped drinking, the majority described how they had “reduced drinking” and “now limited (their) alcohol intake” but had not stopped drinking completely. Most participants who reported reducing their alcohol consumption described health and social benefits associated with this alcohol reduction including improvements to their family and romantic relationships, engagement in work and occupational activities, finances, and health. One participant spoke of how reducing his alcohol intake had improved his health and financial situation:
I have reduced my alcohol and I no longer drink a lot. I used to drink Monday to Monday, now alcohol no longer controls me. I no longer get sick easily, I am healthy, I am alright. I can do my budget and see what we are short of at home. Before I used to just drink money. I can afford a lot of things now because I no longer waste money like before.(Male PID 72; 6MFU)

Despite the perceived benefits of alcohol reduction, several participants still viewed alcohol cessation as an important goal. They described wanting to “totally quit” but finding it “difficult and quite challenging to stop” drinking alcohol. For these participants, reducing their alcohol intake was an incremental step towards their goal of abstinence.

I have started to reduce. Sometimes I drink, but not too much. Eventually I want to stop drinking altogether, as times goes on. (Female PID 9; post-intervention)

You don’t just easily stop drinking, you can’t say you will stop tomorrow. I am trying to reduce my alcohol intake. (Male PID 17; post-intervention)

### 3.3. Suggestions for Modifications to the Intervention

Participants made several suggestions for how to improve the reach and impact of the intervention. These recommendations relate primarily to the content, delivery, and dosage of the intervention. In terms of content, some participants requested more detail about how alcohol affects HIV disease progression (in addition to information on how it affects ART adherence). These participants wanted to “know exactly what alcohol does to the body”. Furthermore, the intervention focused on promoting alcohol reduction and provides little guidance on the benefits of alcohol abstinence. Several of the participants who wanted to stop drinking thought the intervention should be expanded to include more information on alcohol cessation:
They can teach you more in terms of how you can stop instead of only how you can reduce. (Male PID 70; 6MFU)

Some participants also requested additional supplementary material that addressed some of the structural drivers of alcohol use in their context, such as unemployment and a lack of work and income generation skills. These participants suggested supplementing the individual behavior change intervention to include additional components focused on developing income generation skills and work preparedness:
You should try to help find jobs for us. Help us find jobs so that the stress that we have can be reduced… or projects so that even where are no jobs we have something to hold on to. (Female PID 4; post-intervention)

Only a few participants made recommendations for modifications to the delivery of the intervention. Hardly any participants reported barriers to attending intervention sessions at the health facility. A few did mention initial difficulties in taking time off work to attend these sessions but described how these barriers were addressed by the counsellors being able to accommodate them on weekends. Some participants mentioned that they would have preferred to have received counselling at their homes or in their communities rather than at the facilities. These were mainly men who were concerned about HIV-related stigma and did not want to be seen frequenting the health facility:
Most people have a problem of stigma, especially men we don’t even come to the clinic. So stigma is the main problem for men, there are a lot of people who have it. If you did house to house, it was something else… if you did house to house you will see a lot who have secrets. (Male PID 58; 6MFU)

When asked about whether the dosage of the intervention was sufficient to meet their counselling needs, a few participations considered four sessions delivered over two days adequate to meet their counselling expectations. This was particularly the case for those who reported low risk drinking. However, participants who reported excessive alcohol use expressed interest in receiving additional sessions to support their efforts to change and to help them stop drinking completely:
I would have liked to have more sessions and then maybe I would be able to take out what is in my heart, because at least there is someone who I can talk to. I can say they have helped me because I was able to see my problems, but for now I have not yet found a solution… we only had two sessions, so in those two sessions I can’t just make a decision. (Female PID 7; post-intervention)

Some of the participants who expressed interest in receiving additional sessions thought that these sessions could be offered as optional booster sessions that could be accessed on an as-needed basis, rather than making these additional sessions a mandatory part of the intervention package. According to these participants, this would provide them with opportunities to contact their counsellor when difficult problems arise for which they need additional support. As one participant commented:
Sometimes I would come across problems and I wouldn’t know how to solve them, so if I could be able to come here and talk to her so that she can help me. (Female PID 61; 6MFU)

## 4. Discussion

This study provides qualitative evidence of the perceived acceptability and usefulness of a brief MI-PST intervention for facilitating changes to alcohol use among PLWH in South Africa. More specifically, findings suggest that (i) it is generally acceptable to offer PLWH an intervention to address excessive alcohol use, with no notable differences found between men and women; (ii) participants thought the MI-PST intervention helped them reduce their alcohol consumption; (iii) enhancing problem- and emotion-focused coping skills seemed to support participants’ efforts to reduce their alcohol intake; and (iv) minor modifications to the dosage, content and delivery of the intervention could potentially enhance its acceptability.

Given high levels of hazardous alcohol use in their communities and some recognition of the effect that alcohol use has on health and response to HIV treatment, most participants reported that it was appropriate and acceptable to provide alcohol-focused interventions to PLWH, such as themselves. Participants described the counselling as a useful adjunct to their HIV clinical care where little attention was given to psychosocial factors that impact on physical well-being. The use of trained counsellors who were not their usual HIV care providers seemed to enhance the acceptability of screening and the brief alcohol reduction intervention. Professional counsellors are, however, absent from the current public health workforce, largely due to the country’s chronic shortage of financial and human resources for mental health [32,33]. Implementing screening and brief alcohol reduction interventions on scale, therefore, will require alternative delivery agents. Although not examined here, other South African studies have demonstrated the feasibility, acceptability, and effectiveness of using community health workers to conduct screening and brief alcohol reduction interventions [27,28,29]—provided these non-professional providers have adequate training and support.

Despite alcohol-focused screening and interventions being broadly acceptable, some participants acknowledged feeling ambivalent about being screened. Stigma associated with problematic alcohol use for PLWH and limited literacy around alcohol-related issues seemed to contribute to this ambivalence. This observation is in keeping with findings from other South African studies [20,34]. As some participants expressed concern about the impact of these barriers on the uptake of screening and counselling for alcohol problems, any efforts to scale up the provision of these services should be accompanied by activities aimed at reducing health worker stigma towards PLWH who drink and improving patients’ health literacy. National implementation of the recently developed health promotion tool kit, which provides psychoeducation about alcohol (and other) risks for ill-health for patients attending primary care services [35], may assist in raising alcohol-related health literacy.

In addition, findings suggest that the intervention assisted participants in reducing their alcohol intake, with almost all participants describing how they had limited their alcohol use since receiving the intervention. Their descriptions of the intervention’s benefits are supported by findings that these participants significantly reduced the average number of alcoholic drinks consumed per month from 28 at baseline to 9 at the study’s six-month endpoint. Many of the participants reported being unaware that they were drinking at potentially hazardous levels. They described how the personalized feedback they received about their risks associated with their level of alcohol use surprised and motivated them to reduce their drinking. Aspects of the intervention that provided psychoeducation about standard drinks and container sizes and tools to help them track the volume of alcohol consumed seemed particularly helpful for supporting participants’ efforts to change. In addition, participants seemed to value the new problem-solving skills that they had gained through the intervention. They felt the intervention provided them with practical tools for managing life stressors, regulating negative emotions, and accepting and managing their HIV diagnosis. These were salient issues for men and women in this study. Many participants provided concrete examples of how they had applied these skills to cope with problems where previously they used alcohol as a form of avoidant coping.

Despite the perceived benefits of the intervention, almost all participants thought that the usefulness of the intervention could have been enhanced through providing opportunities for additional counselling sessions. Participants who reported alcohol cessation as their behavioral change goal reflected that the intervention was helpful for alcohol reduction efforts but not sufficient for cessation goals. In previous studies using MI-PST, the counselling sessions were delivered over a four to six-week period and comprised four separate counselling occasions, each spaced at least a week apart (see [25,26,27,28,29]). This gave participants multiple opportunities to put their problem-solving skills into practice and to review these efforts with their counsellor at their next session. In the current study, the intervention content was delivered over two occasions, primarily due to initial concerns about the feasibility of retaining participants in an intervention spread over four weeks. While PLWH in South Africa have reported structural barriers to retention in alcohol counselling [36,37], and providing fewer but longer and more intensive sessions is one way of addressing these barriers, structuring the intervention in this way arguably provided participants with fewer opportunities to test their problem-solving skills and review these with their counsellor. Lengthier counselling sessions may also raise concerns about the feasibility of implementing this intervention at scale with high patient caseloads generally limiting the amount of time health workers are able to spend with patients [38]. Given that several participants seemed to want additional contact points with their counsellor, future applications of this intervention should consider reverting to delivering the intervention modules over more than two contact sessions. This will reduce the length of the counselling sessions and may make these sessions more feasible to deliver in busy health care settings.

To address concerns about the addition of counselling sessions and risk of attrition among individuals who experience barriers in attending facility-based services, future applications of MI-PST could consider offering patients alternatives to face-to-face counselling at health care facilities. In keeping with other studies of PLWH [20,39], some participants in this study (mainly men) raised concerns about receiving behavioral intervention at health facilities due to difficulties in taking time off work and concerns about stigma. These concerns could be partially mitigated by offering alternatives to facility-based counselling. One option is telephone counselling. Although not widely used in low-and-middle-income countries, there is evidence from high-income countries of the effectiveness of telephone-based MI [40] and PST [41,42]. Another option is the provision of counselling at patients’ homes or other community settings. As community health workers already provide adherence support and care to PLWH in their homes and other community settings, this may be a feasible option. Additional studies are needed to explore the feasibility and acceptability of community-health worker-delivered alcohol reduction interventions in each of these settings. 

Similar to other studies of psychosocial interventions in this setting [43,44,45,46], several participants also expressed the need for ongoing support for change either through community-based services (including home visits) or continued access to a facility-based psychosocial counsellor. To address this unmet need, future studies should consider expanding the intervention to include extended support for change. Given that hazardous drinking is normative among those who drink in many parts of South Africa, including Tshwane [10], this modification may be necessary to help participants sustain the initial reductions that they made to their alcohol use.

In addition to modifying the format and mode of delivery, several participants suggested supplementing the intervention with additional services to help address some of the contextual factors that contribute to heavy drinking, such as lack of employment and income generation opportunities which was a major source of stress. Furthermore, as many of the problems that participants reported seemed to stem from conflict in relationships and emotional regulation difficulties, additional content that focuses specifically on emotional regulation and conflict resolution skills may increase the impact of this intervention.

There are some study limitations that should be considered when interpreting these findings. While we took precautions to limit social desirability bias, participants may not have felt comfortable criticizing the intervention. Second, we acknowledge that study findings may not be generalizable to other populations of patients receiving ART—although this is not the purpose of qualitative research. However, our findings provide support for the further study of this intervention in other patient populations and contexts. Third, as the intervention was delivered by research staff, counselling could be offered outside of the usual operating hours of health facilities (including over weekends) and barriers to access minimized. This may not have been possible if the intervention had been delivered by usual care providers. A pragmatic evaluation of the intervention is still needed to determine whether patient time, transport and cost barriers impact on the feasibility of implementing this intervention outside of the context of a randomized trial.

## 5. Conclusions

In conclusion, this study provides qualitative evidence, from a patient perspective, of the acceptability and usefulness of a brief MI-PST intervention for reducing alcohol consumption among PLWH. Findings highlight the desire and need for psychosocial interventions that allow PLWH to talk about what matters most to them. Findings also demonstrate the value of providing PLWH with personalized feedback on the health risks associated with their alcohol consumption patterns and helping PLWH develop non-alcohol related strategies for coping with life problems and emotional stress as a means of facilitating sustained changes to alcohol consumption. While there are unanswered questions about how best to format and structure the intervention for feasibility of implementation in usual HIV services, our findings indicate the acceptability and usefulness of the intervention content. This provides support for adaptation and further study of the implementation of the intervention in usual care settings.

## Figures and Tables

**Table 1 ijerph-17-05706-t001:** Overview of intervention sessions and components.

Session	Module	Activities
Session 1	Module 1	Screening of alcohol use Provide feedback on results of screening Increase knowledge of how alcohol use impacts on course of HIV Assess readiness to change Review pros and cons of change Goal setting
Session 1	Module 2	Explain the link between problems and alcohol use, and the rationale for Problem solving therapy (PST) Identifying things that are important to the participant Describe the steps of PST First Problem Solving Session with counsellor (using the steps) and describe homework
Session 2	Module 3	Patient check-in (using MI) Review homework from previous week Review PST steps and affirm attempts to change Explain what can be done about problems that are not important (coping with negative thoughts) Second Problem Solving Session with counsellor and an exercise
Session 2	Module 4	Explain what can be done about problems that are important but cannot be solved (advancing the process of acceptance) Third Problem Solving Session with counsellor Summary

**Table 2 ijerph-17-05706-t002:** Demographic and clinical characteristics of participants.

Characteristics	Selected, not Interviewed (*n* = 24)	Selected, and Interviewed (*n* = 49)	*p* Value
Male (%, *n*)	33.3% (8)	32.7% (16)	0.95
Age (M, SD)	38.3 (7.5)	41.1 (9.3)	0.21
Completed high school (%, *n*)	37.5% (9)	38.8% (19)	0.45
Unemployed (%, *n*)	41.7% (10)	44.9% (22)	0.39
Income less than R1600/month * (%, *n*)	50.0% (12)	51.0% (25)	0.47
AUDIT score: M (SD)	9.1 (5.2)	8.6 (4.6)	0.66
Above cut-off for probable depression ** (%, *n*)	70.8% (17)	75.5% (37)	0.67

* ZAR1600~USD 150 ** The Center for Epidemiology Scale on Depression was used to screen for depression.

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
