# Peer review of "“Moving Forward with Life”: Acceptability of a Brief Alcohol Reduction Intervention for People Receiving Antiretroviral Therapy in South Africa"

_ijerph, 2020, doi:10.3390/ijerph17165706_

Round 1
Reviewer 1 Report
This is a well conducted and described study. I have relatively minor suggestions to improve the manuscript, described below.
ABSTRACT: The authors describe key themes that emerged from their qualitative interviews, however those themes are vague and do not support the conclusions presented on their abstract. Please review. This is NOT the first study to describe how people living with HIV/AIDS apply learned skills at an intervention to cope with stressors - many studies addressing addiction, alcohol use/abuse/dependency, psychological distress among many others aspects that might influence HAART adherence had already described this.
INTRODUCTION
1st paragraph: Please add the current prevalence of HIV in SA, and the current scenario for HIV cascade in the country - comparing it to UNAIDS 90-90-90 goals.
2nd paragraph: Please add the current prevalence of alcohol consumption in SA.
3rd & 4th paragraph: Those paragraphs actually describe their methods. The authors could moved it to the next section.
It would be interesting to compare the epidemiological context of SA with other African countries and other regions - this could highlight the impact of both HIV/AIDS epidemic and alcohol consumption in SA.
METHODS: Please consider including a figure with the framework utilized to conduct your qualitative analysis.
RESULTS: Please include a Table with basic SES info from study participants and a brief description of their characteristics.
DISCUSSION: South Africa has over 7 million PLWHA, a population with high prevalence of alcohol/substance abuse and dependency and mental disorders. Yet, South Africa’s mental healthcare resources are wholly unequipped to handle this burden. It is important to address the country reality, and include feasible recommendations outside the context of a RCT. How could we address this key public health problem (alcohol use/abuse/dependency among PLWHA) with the magnitude of the HIV/AIDS pandemic in South Africa and the available resources? Is it possible to scale up this intervention, training community leaders, educators etc?
CONCLUSIONS
Again, this is NOT the first study to provide feedback on how study participants apply learned skills during an intervention to manage their life struggles. This has been implemented and evaluated by several groups from South Africa and abroad, addressing a broad range of public health problems - including HIV/AIDS and alcohol use/abuse/dependency. See, for instance, intervention implemented by MRC South Africa, studies conducted by Dr. Diana Huis in ’t Veld and colleagues, Dr. Munyaradzi Madhombiro, Dr. Sarah Gordon, Dr. Seth Kalichman…(this is a non-exhaustive list)
Thanks for the opportunity to review your manuscript.
Author Response
Thank you for these thoughtful comments. We have responded to these and believe the changes that we have made strengthen the paper. Below we describe our responses to each of the points raised, in italics:
- ABSTRACT: The authors describe key themes that emerged from their qualitative interviews, however those themes are vague and do not support the conclusions presented on their abstract. Please review. We have made changes to our description of the key findings in the abstract
- This is NOT the first study to describe how people living with HIV/AIDS apply learned skills at an intervention to cope with stressors - This sentence has been removed from the abstract
- INTRODUCTION: 1st paragraph: Please add the current prevalence of HIV in SA, and the current scenario for HIV cascade in the country - comparing it to UNAIDS 90-90-90 goals. Thank you for this suggestion- this has been attended to
- 2nd paragraph: Please add the current prevalence of alcohol consumption in SA. This has now been added to the Introduction
- 3rd & 4th paragraph: Those paragraphs actually describe their methods. The authors could moved it to the next section. We have integrated this information into the section of the methods that describes this intervention
- It would be interesting to compare the epidemiological context of SA with other African countries and other regions - this could highlight the impact of both HIV/AIDS epidemic and alcohol consumption in SA. We agree that it could be interesting- we have added some information to the introduction that describes the syndemic of alcohol and HIV in SA and other African countries. We have decided to limit the extent to which we describe the epidemiology given the focus of the paper
7. METHODS: Please consider including a figure with the framework utilized to conduct your qualitative analysis. We have clarified that this was a coding framework rather than a theoretical framework. In the interest of space and brevity, we have chosen not to include this framework in the paper as it is not part of the COREQ guidelines for reporting qualitative research.
8. RESULTS: Please include a Table with basic SES info from study participants and a brief description of their characteristics. We have moved Table 2 from the methods section to the results and expanded it to include additional information on education, employment and income
9. DISCUSSION: It is important to address the country reality, and include feasible recommendations outside the context of an RCT. This point is well-taken. As far as possible we have integrated recommendations for adapting the intervention for implementation on scale and how implementation of the intervention may work in practice given the realities of South Africa's resource constrained health system.
CONCLUSIONS: Again, this is NOT the first study to provide feedback on how study participants apply learned skills during an intervention to manage their life struggles. Thank you for this- we have removed the sentence from the conclusion have re-written it in line with the other reviewer's comments.
Reviewer 2 Report
Thank you for allowing me to review this most excellent paper. The subject matter is relevant and novel, the study appears to be very sound and the findings are robust. I have only a few critical comments for this otherwise impressive paper.
First, lines 433-436 indicates that generalizability beyond the sample is a limitation, however, the purpose of qualitative data is not to generalize but to elucidate the lived experience of participants. I suggest rewording this to indicate that your unique findings provide solid support for further study that may be generalizable beyond this sample. Also, you may want to eliminate the phrase "who participated" on line 433 as it seems to be redundant.
Secondly, lines 437-441 seems very speculative as to why some eligible people chose not to participate. This is not a limitation in qualitative research as, again, you are seeking to tell the story of the participants. Any alteration in your sample would yield different results. You are not wanting to generalize findings, so selection bias, etc. are not limitations.
Finally, your conclusion seems a bit apologetic for the non-generalizability of the study findings. You have a wonderful qualitative study that identified some very important points, don't apologize that your "apple is not an orange". Try to present your findings in a more positive manner that presents your findings and tells the reader that they are the foundation for more research. I also think a key finding in your study was the strong desire and need for psychosocial interventions with this community. They seem to be telling you that more than anything they want to talk about the things that matter most to them (emotions, feelings, etc.).
line 47: missing word
Again, thank you for allowing me to review this paper, it really is very well done!
Author Response
Thank you for the very positive comments and review of this manuscript. We have considered and attended to the concerns raised and these are described below in italics.
hank you for allowing me to review this most excellent paper. The subject matter is relevant and novel, the study appears to be very sound and the findings are robust. I have only a few critical comments for this otherwise impressive paper.
- First, lines 433-436 indicates that generalizability beyond the sample is a limitation, however, the purpose of qualitative data is not to generalize but to elucidate the lived experience of participants. This is noted and we have revised this limitation accordingly
- I suggest rewording this to indicate that your unique findings provide solid support for further study that may be generalizable beyond this sample. Also, you may want to eliminate the phrase "who participated" on line 433 as it seems to be redundant. This section has been revised
- Secondly, lines 437-441 seems very speculative as to why some eligible people chose not to participate. This is not a limitation in qualitative research as, again, you are seeking to tell the story of the participants. Any alteration in your sample would yield different results. You are not wanting to generalize findings, so selection bias, etc. are not limitations. This point is well-taken and we have removed this sentence from the section on limitations.
- Finally, your conclusion seems a bit apologetic for the non-generalizability of the study findings. You have a wonderful qualitative study that identified some very important points, don't apologize that your "apple is not an orange". Try to present your findings in a more positive manner that presents your findings and tells the reader that they are the foundation for more research. I also think a key finding in your study was the strong desire and need for psychosocial interventions with this community. They seem to be telling you that more than anything they want to talk about the things that matter most to them (emotions, feelings, etc.). On re-reading the conclusion, we agree that we did seem apologetic. We have rewritten aspects of the conclusion to present the findings in a more positive light.
- line 47: missing word. Thank you for picking this up- it has been corrected.